# Contribution of Sorting Nexin 3 in the Cytomegalovirus Assembly

**DOI:** 10.3390/biomedicines13040936

**Published:** 2025-04-11

**Authors:** Ivona Viduka, Igor Štimac, Silvija Lukanović Jurić, Tamara Gulić, Berislav Lisnić, Gordana Blagojević Zagorac, Hana Mahmutefendić Lučin, Pero Lučin

**Affiliations:** 1Department of Physiology, Immunology and Pathophysiology, Faculty of Medicine, University of Rijeka, Braće Branchetta 20, 51000 Rijeka, Croatia; ivona.viduka@uniri.hr (I.V.); igor.stimac@uniri.hr (I.Š.); silvijalj@uniri.hr (S.L.J.); tamara.gulic@uniri.hr (T.G.); gordana.blagojevic@uniri.hr (G.B.Z.); hana.mahmutefendic@uniri.hr (H.M.L.); 2Center for Proteomics, Faculty of Medicine, University of Rijeka, Braće Branchetta 20, 51000 Rijeka, Croatia; berislav.lisnic@uniri.hr; 3Campus University Center Varaždin, University North, Jurja Križanića 31b, 42000 Varaždin, Croatia

**Keywords:** cytomegalovirus, assembly compartment, beta-herpesvirus secondary envelopment, sorting nexin 3, sorting nexin 27, Rab10, tubular recycling endosomes

## Abstract

**Background/Objectives**: Cytomegalovirus (CMV) infection expands early endosomes (EEs) into tubular extensions that may contribute to the control of virus replication and virion assembly. Sequential recruitment of protein coats and sorting nexins (SNXs) creates membrane zones at the EEs that serve as scaffolds for membrane tubulation and retrieval of cargo proteins, including host cell signaling proteins and viral glycoproteins. This study aims to investigate whether the SNX3-dependent zone of EEs contributes to CMV replication and assembly. **Methods**: Protein localization was analyzed by confocal imaging and expression by Western blot. The contribution of SNX3 to murine CMV (MCMV) replication, assembly compartment (AC) formation, and virion release was analyzed by siRNA and shRNA depletion. The impact of other downstream SNXs that act in EE tubulation was investigated by combined siRNA knockdowns of SNX1, SNX2, SNX4, SNX17, and SNX27 on cell lines expressing shRNA for SNX3. **Results**: The SNX3-162 isoform acting at EEs was efficiently knocked down by siRNA and shRNA. The SNX3-dependent EE zone recruited SNX27 and contributed to Rab10-dependent tubulation within the pre-AC. SNX3 was not essential for MCMV replication but contributed to the SNX27-, SNX17- and SNX4-dependent release of virions. Silencing SNX3 further reduced the release of virions after silencing SNX27, SNX4, and SNX17, three SNXs that control recycling to the plasma membrane. **Conclusions**: SNX3 contributes to the formation of pre-AC and MCMV assembly. It acts sequentially with SNX27, SNX4, and SNX17 along the recycling pathway in the process of the production and release of infection virions, suggesting that multiple membrane sources may contribute to the secondary envelopment of MCMV virions.

## 1. Introduction

Cytomegaloviruses (CMVs), like other beta-herpesviruses, infect a large proportion of the human population [1]. Infections are usually asymptomatic, as the immune system effectively combats viral replication [2], with severe clinical manifestations in immunocompromised or immunologically immature individuals [3]. However, the immune system is unable to eliminate the virus and CMV remains in the body in a state of latency and reactivates with unknown frequency [1]. Reactivation is thought to be associated with many pathophysiological conditions, including severe infections when reactivated in immunocompromised individuals [1,2]. Such a complex interaction with their hosts is mainly related to the complexity of their coding potential, which is the largest among viruses, with almost 200 genes and more than 800 coding variants [4,5,6]. This coding potential has evolved with the hosts over thousands of years and has enabled complex interactions with the physiological processes of the host, both at the cellular level and at the level of the whole organism. These evolutionary developments are of great importance from the perspective of studying cell physiology, as CMVs have a long time frame to identify and develop gene products that can target critical points in these processes. However, the complexity of the interactions complicates the organization of studies on CMV biology and pathogenesis. Although human CMV (HCMV), as the major human pathogen, is the focus of research efforts, studies are also being conducted with animal CMVs, particularly mouse CMV (MCMV), which allows testing in animal models to overcome the complexity [7]. Experience to date suggests that many functions are conserved among beta-herpesviruses, and progress in understanding CMV biology may require the compilation of data from MCMV infections as well.

Infection with CMV leads to a complete restructuring of the entire cell and the formation of two megastructures, the nuclear replication centers and the cytoplasmic assembly complex (AC) [8,9]. The AC comprises a large area, almost as large as the nucleus, containing displaced Golgi and expanded endosomal systems that, together, form the basic configuration of the AC [10,11,12,13]. This conspicuous structure rises from the pericentriolar region and expels much of the endoplasmic reticulum, late endosomes, and secretory organelles toward the cell periphery [10,13]. In HCMV-infected cells, the establishment of the AC requires the expression of several HCMV-encoded late (L) gene products, after replication of viral DNA [10,11,12], and takes 2–3 days [10,11,12]. In contrast, in MCMV-infected cells, the basic configuration of the AC is rapidly established 5–7 h post-infection (hpi) by the activation of early (E) gene products, and does not require viral DNA replication [13,14,15,16]. In the context of MCMV infection of fibroblast-like cells, the E phase of infection covers the period from 1 hpi to 15–16 hpi, the time of onset of viral DNA replication, which is followed by the late phase (L) [6]. The basic configuration of the AC which is established in the E phase is referred to as the pre-AC [13]. The topology of the AC is similar in HCMV- and MCMV-infected cells [11,13], suggesting that the underlying mechanisms are conserved between beta-herpesviruses. Although several HCMV gene products encoding tegument proteins [12] and microRNAs [17] have been identified, the mechanism of intervention in host cell processes and AC biogenesis is unknown. Its rapid development in MCMV-infected cells prior to the expression of nearly two-thirds of the MCMV coding potential, which occurs after viral DNA synthesis [6] and colonizes the AC structure, facilitates the development of experimental approaches that could reveal the mechanisms of AC biogenesis [13].

In our previous studies [13,15,16,18,19], we have shown that the basic configuration of the AC arises from a rapid reorganization of the Golgi into a ring-like structure around expanded membrane domains of early endosomes (EEs), recycling endosomes (REs)/the endosomal recycling compartment (ERC) and the trans-Golgi network (TGN). These events occur simultaneously, and it is not known whether they are coupled. The expansion is reflected in an over-recruitment of many host cell proteins that regulate membrane flux in these compartments and their intermediates [13]. These proteins oscillate between the cytosolic and membrane-associated states and, when associated with membranes, determine the functional properties of a membrane domain. Many of the proteins that accumulate at membranes in the AC are barely detectable in non-infected cell membrane systems, consistent with their rapid turnover and high domain dynamics. Their retention at membranes within the AC indicates an expansion of membrane domains at the EE-RE/ERC-TGN interface, much slower dynamics (i.e., inhibited endosomal recycling), and the establishment of an organelle structure that is significantly different from that of uninfected cells and involves a reorganization of membrane fluxes and pathways [13,16,20].

The distinguishing feature of the pre-AC is the expansion of tubular recycling endosomal (TRE) domains and tubular domains of the ERC [13,14], which is reflected in the increased membrane recruitment of proteins that regulate membrane tubulation [13]. The excessive recruitment of ADP ribosylation factor (ARF) proteins of all classes and high levels of colocalization with internalized transferrin receptors (TfRs) indicate tubulation from EEs [19], such as Arf1- and Arf3-dependent tubulation from the Rab4-dependent domain for endosomal recycling of clathrin-dependent endocytic (CDE) cargo [21]. Excessive recruitment of Rab10 is associated with the tubulation of EE membranes after sorting clathrin-independent endocytic (CIE) cargo [22]. In our recent study, we have shown that Rab10 expansion requires a sorting mechanism involving retromer, sorting nexin (SNX) 27, and endosomal sorting complex for promoting exit 1 (ESCPE-1) [18]. These complexes are sequentially assembled on EE membranes, retrieve CIE cargo, and initiate membrane bending. The bent membranes, with the retrieved CIE cargo, recruit EHBP1 and Rab10, which are required for membrane growth and convert the bend into a tubule [23]. Further recruitment of MICAL-L1 enhances tubule growth and is required for tubule fission into recycling carriers. In non-infected fibroblasts, these processes occur rapidly, and Rab10-positive domains (Rab10-PDs) are short-lived, resulting in a low association of Rab10 with membranes [24]. In MCMV-infected cells, tubulation termination is altered and Rab10-PD expands, leading to accumulation within the pre-AC [23].

Sorting nexins play a seminal role in the biogenesis of tubular endosomes. They regulate cargo sorting events at EEs [25], which are organized at distinct zones at EE membranes [26]. Membrane zones within EEs can be induced by proteins that contain phox (PX) or FYVE domains, canonical readers of phosphatidylinositol 3-phosphate (PI3P), a major component of the lipid codon at EEs [27]. SNXs contain a PX domain that binds mainly PI3P at EE membranes, and thereby they seed the formation of multimeric assemblies that reshape the membrane and attract cargo for sorting and recycling to the plasma membrane (PM) or to the TGN [25,26]. For example, SNX1 forms a zone which recruits SNX27, which binds retromer for CIE cargo sorting to the PM [28], whereas SNX3 forms a zone and recruits retromer for sorting several cargo proteins to the TGN [29]. SNX3 can also engage SNX27 [28,30,31] and sort CIE cargo into the Arf6-dependent recycling route to the PM independently of retromer [32], but it is unknown whether these events are coupled. Some SNXs, such as SNX1 and SNX2, contain a Bin1/Amphyphysin/Rvs (BAR) domain that enables multimerization and the induction of membrane curvatures required for tubulation [25], whereas some SNXs, such as SNX3, can induce tubulation by inserting their PX domain into curved membranes [33]. SNXs without and with a BAR domain segregate to earlier and later parts of EEs, respectively [34,35]. SNX3 is enriched in a subdomain of EEs [35] and competes with EEA1(Early endosome antigen1) and Hrs (hepatocyte growth factor-regulated tyrosine kinase substrate) [32,36], two major FYVE domain-containing PI3P binding regulators [37].

Considering that SNX3 also contributes to the retrieval of CIE cargo [32] and engages SNX27 [28,30,31], it could contribute to the early events in the formation of the AC. Therefore, the aim of this study was to investigate whether the SNX3-dependent pathway contributes to the expansion of Rab10-PD, pre-AC biogenesis, and infectious virion production during infection with MCMV.

## 2. Materials and Methods

### 2.1. Cell Lines

The fibroblast-like cell line NIH3T3 (American Type Culture Collection, Manassas, Virginia, USA; ATCC CRL-1658) was used for most experiments, and in part Balb3T3 (ATCC clone A31, CCL-163). NIH3T3-pEGFP-mSNX27 [18] and NIH3T3-pEGFP-Rab10 [23], cell lines with doxycycline-inducible expression of EGFP-mSNX27 and EGFP-Rab10, respectively, were used for colocalization analysis. Primary murine embryonic fibroblasts (MEFs) from 17-day-old Balb/c mouse embryos were used for virus production and plaque assay. Cells were cultured in Dulbecco’s Modified Eagle’s Medium (DMEM) supplemented with 10% (5% for MEFs) fetal bovine serum (FBS) containing 2 mM L-glutamine, 100 mg/mL streptomycin, and 100 U/mL penicillin (all reagents from Gibco/Invitrogen, Grand Island, NY, USA) at 37 °C and 5% CO_2_.

### 2.2. Viruses and Infection Conditions

To avoid FcR-mediated non-specific binding of antibody reagents, the recombinant virus ∆m138-MCMV (∆MC95.15) with the deletion of the fcr1 (m138) gene [38] was used for the experiments. The growth characteristics do not differ from those of the wild-type virus [38], including the establishment of the AC [13,15,16,18,23,39]. A recombinant virus C3X-GFP-MCMV (MCMV-GFP) expressing green fluorescent protein (GFP) from the immediate–early (IE) phase of infection [40] was used to monitor MCMV replication by flow cytometry. To track MCMV capsids in the L phase of infection, we used recombinant S-mCherry-SCP-MCMV with fluorescently labeled small capsid protein (SCP) [41]. Virus stocks were prepared according to standard procedures, and cells were infected with 1 PFU/cell by centrifugation to increase infectivity at a multiplicity of infection (MOI) of 10 [42]. In brief, viral stocks were produced by infecting mouse embryonic fibroblasts (MEFs) with MCMV at a dose of 0.01 PFU per cell followed by incubation at 37 °C for approximately 5 days, or until the cells had completely rounded up. The medium in which the cells had grown was collected and centrifuged at 500× *g* for 10 min. The obtained supernatant was further ultracentrifuged (Thermo Fisher Scientific, Waltham, MA USA) at 50,000× *g* for 90 min at 4 °C to obtain the viral pellet. This pellet was purified from residual cellular debris by ultracentrifugation in a density gradient (15% sucrose; 70,000× *g*, 90 min at 4 °C) and dissolved in sterile PBS. The resulting suspension was aliquoted (25 μL) and stored at −80 °C. The virus titer was determined using a viral plaque assay. As previously reported [13,16], infection efficiency was assessed by immunofluorescence detection of pIE1, and infection efficiency of 96% achieved at an MOI of 10 was used for immunofluorescence studies.

### 2.3. Antibodies and Reagents

Antibodies against markers of endosomal compartments were monoclonal (mAb) or polyclonal (pAb), as follows: rabbit pAb against SNX3 (Cat. No. 10772-1-AP) and mouse IgG1 mAb against Rab11A (Cat. No. 67902-1-Ig) both from Proteintech, Rosemont, IL, USA; rabbit mAbs against Rab10 (Cat. No. 8127) and EEA1 (Cat. No. 3288), and mouse IgG1 mAb against Rab5A (Cat. No. 46449) from Cell Signaling Inc., Danvers, MA, USA; mouse IgG_2b_ mAb against Vps35 (Cat. No. sc-374372; Santa Cruz Biotechnology, Dallas, TX, USA); mouse IgG_1_ mAb against GM130 (Cat. No. 610823) and rat IgG_2a_ mAb against Lamp1 (Cat. No. 553792) from BD Biosciences, Franklin Lakes, NJ, USA; mouse IgG_1_ mAb against actin (Cat. No. MAB1501; Millipore, Burlington, MA, USA); and rat IgG mAb against transferrin receptor (TfR, clone R17 217.1.3; ATCC TIB 219).

Anti-MCMV mAbs were produced and verified by the Center for Proteomics of the University of Rijeka (https://products.capri.com.hr/shop/?swoof=1&pa_reactivity=murine-cytomegalovirus; accessed on 15 December 2024). We used the following: mouse mAbs IgG_1_ (clone CROMA101) and IgG_2a_ (clone IE1.01) against pm123/pIE1, mouse mAbs IgG_1_ (clone M55.01 for Western blot) and IgG_2a_ (clone M55.02 for immunofluorescence and FACS analysis) against pM55/gB, mouse mAb against pM57 (clone M57.02), mouse mAb IgG_1_ against pm74 (clone 74.01), and mouse mAb IgG_1_ (clone M25C.01) against pM25.

The following secondary antibodies were used for immunofluorescence or FACS analysis: Alexa Fluor (AF) 488-, 555-, 594-conjugated (Molecular Probes; Leiden, The Netherlands) and AF680-conjugated (Jackson ImmunoResearch, West Grove, USA) antibodies against mouse IgG_1_, mouse IgG_2a_, mouse IgG_2b_, and rabbit Ig, respectively. Western blot analysis was performed with goat anti-rabbit and goat anti-mouse antibodies conjugated to HRP (horseradish peroxidase) (Jackson ImmunoResearch, West Grove, PA, USA).

Puromycin was purchased from Santa Cruz Biotechnology Inc. (Dallas, TX, USA) and DAPI (4,6-diamidino-2-phenylindole dihydrochloride) was purchased from Thermo Fisher Scientific (Cat. No. D1306; Waltham, MA, USA). Sigma-Aldrich Chemie GmbH (Schnelldorf, Germany) provided propidium iodide and other chemicals.

### 2.4. Immunofluorescence and Confocal Microscopy

The 60–70% confluent cells were seeded on coverslips in 24-well plates for experiments based on immunofluorescence analysis, to minimize heterogeneity [43]. After fixation with 4% paraformaldehyde (PFA) for 20 min at room temperature (r.t.) and permeabilization with 1% Tween 20 (20 min at 37 °C), the cells were incubated with primary antibodies for 60 min at r.t., washed 3 times with PBS and incubated with the corresponding fluorochrome-conjugated secondary antibodies for another 60 min at r.t. Cells were rinsed in PBS, embedded in Mowiol (Fluka Chemicals, Selzee, Germany)-DABCO (Sigma Chemical Co., Steinheim, Germany) in PBS containing 50% glycerol and analyzed by confocal and/or epifluorescence microscopy.

Imaging was performed using either the Olympus Fluoview FV300 confocal microscope (Olympus Optical Co., Tokyo, Japan) or the Leica DMI8 inverted confocal microscope (Leica Microsystems GmbH, Wetzlar, Germany). The FV300 microscope was equipped with Ar 488, He/Ne 543, and He/Ne 633 lasers, and Fluoview software, version 4.3 FV 300 (Olympus Optical Co., Tokyo, Japan), a PLAPO60xO objective, appropriate barrier filters, and PMT detectors. The Leica DMI8 microscope (confocal part: TCS SP8; Leica Microsystems GmbH, Wetzlar, Germany) was equipped with HC PLAPO CS2 objective (63×1.40 oil), UV (Diode 405), Ar 488, DPSS 561, and He/Ne 633 lasers, and two PMT and two HyD detectors. The images were acquired under controlled parameter settings with a z-series of 0.5 μm and exported in TIFF format. For epifluorescence microscopy, we used the Olympus BX52 microscope with DP72CCD camera (Olympus, Tokyo, Japan) with UPlanFL N 40 ×/0.75 objective and cellSens Standard 1.15 software.

### 2.5. Image Analysis

A concentrated fluorescent signal within an angle of < 90° indicates the presence of AC in infected cells [15]. The percentage of infected cells with developed AC was determined by direct counting of at least 10 fields of view per sample (200–400 cells) under the epifluorescence microscope.

For colocalization and fluorescence intensity, images with a pixel size of 120.37 × 120.37 nm were analyzed using FIJI (ImageJ 1.54f) (https://imagej.net/software/fiji/; accessed on 15 December 2024) and available plugins. Red, green, and blue channels were split and at least 10–15 cells were analyzed in each experiment. Colocalization was quantified by calculating the Manders’ overlap coefficients (M1 and M2) and the Pearson’s coefficient of the entire z-stack (8–12 confocal slices) using the BIOP JACoP plugin (https://github.com/BIOP/ijp-jacop-b; accessed on 15 December 2024) for 3D analysis [44]. In BIOP JACoP, the Intermodes- or Otsu-algorithm threshold was chosen as an automatic threshold with the defined region of interest (ROI) for each image. Quantification of fluorescence intensity was performed as previously described [45,46]. Briefly, the area, integrated density, and mean gray value were measured after ROIs were selected. The following formula was used to determine the total corrected cell fluorescence (TCCF): integrated density-(area of selected cell x mean fluorescence of background values).

### 2.6. Small Interfering RNA (siRNA)

Small interfering RNA (siRNA) sequences were obtained as follows: non-targeting negative control siRNA (Cat. No. 1022076) and Mm_Snx27_7 sequence (CatNo. SI04939543) were obtained from Qiagen (Hilden, Germany), while siRNAs for SNX1 (sc-41346), SNX2 (sc-41350), SNX3 (sc-41352), SNX4 (sc-41354), and SNX17 (sc-61588) were obtained from Santa Cruz Biotechnology Inc. (Dallas, TX, USA). Reverse transfection with siRNAs and RNAiMAX Lipofectamine Reagent (Cat. No. 13778150; Invitrogen, Carlsbad, CA, USA) was performed according to the manufacturer’s guidelines. When used separately, the final siRNA concentration was 80 nM for SNX1, SNX2, SNX3, SNX4, and SNX17 and 20 nM for SNX27. When combined with siRNAs for SNX1, SNX2, and SNX3, the concentrations of siSNX1 and siSNX2 were 20 nM and 40 nM for SNX3, respectively. For the combination of siSNX3 and siSNX4, 40 nM each was used. The concentrations of siSNX3 and siSNX27 in the combination were 60 nM and 20 nM, respectively. Cells were analyzed 48 h after transfection or infection with MCMV.

### 2.7. Small Hairpin RNA (shRNA) and Cell Line Development

All reagents to produce stably transfected cells with small hairpin RNA (shRNA) were purchased from Santa Cruz Biotechnology Inc. (Dallas, TX, USA). Stable NIH3T3 cell lines were prepared by transfection of control shRNA plasmid-A (sc-108060) which encodes a scrambled shRNA sequence (shScr-expressing NIH3T3 cell line) and the SNX3 shRNA plasmid (sc-41352-SH) for SNX3 silencing (shSNX3-expressing NIH3T3 cell line), using plasmid transfection reagent (sc-108061). Then, 48 h after transfection on a 6-well plate, the medium was replaced with freshly prepared medium containing 2.5 µg/mL puromycin to select stably transfected cells. After 7–10 days, colonies were picked and transferred to a 96-well plate. Confluent wells were further expanded and subjected to Western blot for evaluation of SNX3 expression.

### 2.8. Flow Cytometry

Flow cytometry was performed using a FACSCalibur flow cytometer (Becton Dickinson & Co., San Jose, CA, USA) on 5000/10,000 living cells. Propidium iodide was used for dead cell exclusion.

#### 2.8.1. Detection of Cells Infected with C3X-GFP-MCMV

For monitoring the progression of the E phase of infection, shSNX3- and shScr-expressing fibroblasts were grown in 12-well plates, and the next day were infected with C3X-GFP-MCMV (MOI of 10). At 0, 6, and 24 hpi, samples were collected, and the GFP signal was measured by flow cytometry. The fluorescence signal was calculated as the mean fluorescence intensity (MFI) after subtracting the background fluorescence at 0 hpi (∆MFI).

#### 2.8.2. Cell Surface M55/gB Protein Expression

shScr- and shSNX3-expressing NIH3T3 cells were grown in 12-well plates. The next day, cells were collected (non-infected samples) or infected with ∆MC95.15 (MOI of 10) and harvested at 48 hpi. After a brief trypsinization, cells were incubated at 4 °C for 50 min with primary mAb against pM55/gB (clone M55.02) in FACS buffer (PBS containing 10 mM EDTA, HEPES pH 7.2, 0.1% NaN_3_, and 2% FBS). Cells were washed with FACS buffer and incubated with AF^488^-conjugated secondary antibody against mouse IgG_2a._ After incubation for 40 min at 4 °C, cells were washed and analyzed.

### 2.9. Western Blot

Cells were lysed with RIPA lysis buffer supplemented with protease inhibitors (Pierce RIPA Buffer, Cat. No. 89900; Thermo Fisher Scientific, Waltham, MA, USA) and mixed with sample buffer (50% glycerol, 10% SDS, 0.05% bromophenol blue, 0.3 M Tris, pH 6.8) to obtain whole cell lysates (WCL). After SDS-PAGE (Bio-Rad PowerPac Universal, Hercules, CA, USA), proteins were separated and blotted onto a polyvinylidene difluoride membrane (PVDF-P WB membrane, Millipore, Burlington, MA, USA; Cat. No. IPVH00010) at 80 V for two hours using the Bio-Rad Trans-Blot Turbo Transfer System (Hercules, CA, USA). The membranes were incubated overnight at 4 °C with the appropriate primary antibody after blocking for one hour in 1% blocking reagent (Cat. No. 11921681001; Roche Diagnostics GmbH, Mannheim, Germany). Subsequently, horseradish peroxidase (HRP)-conjugated secondary antibodies were incubated for 60 min at r.t. Both primary and secondary antibodies were diluted in TBS buffer containing 0.5% blocking reagent. T-TBS (TBS with 0.05% Tween 20; pH = 7.5) was used for three cycles of membrane washing between and after antibody incubation.

The signal was detected by chemiluminescence (SignalFire [TM] Plus ECL Reagent or SignalFire [TM] Elite ECL Reagent; Cell Signaling, Cell Signaling Inc., Danvers, MA, USA; Cat. No. 12630S or 12757P, respectively) using ImageQuant LAS 500 (GE Healthcare Bio-Sciences AB, Upsala, Sweden). The protein of interest and β-actin (loading control) were detected on the same membrane. The chemiluminescence signal was quantified using ImageQuantTL software (version 10.2, Cytiva) and normalized to β-actin. The kinetics of protein expression in host cells during MCMV infection as well as changes in MCMV protein expression after siRNA or shRNA treatment were calculated as previously described [18].

### 2.10. RNA Preparation and Real-Time Quantitative Polymerase Chain Reaction (RT-qPCR)

To determine the degree of SNX3 silencing in shRNA cell lines and after siRNA transfection, total RNA was isolated from non-infected and 48 h infected cells using the TRIzol method (Invitrogen, Carlsbad, CA, USA). The quality of RNA samples was analyzed by UV spectrophotometry (high OD_260_/OD_280_ ratio), and high-quality samples (3 μg) were reverse transcribed using a High-Capacity cDNA Reverse Transcription Kit (Applied Biosystems, Foster City, CA, USA, Cat. No. 4368814) according to the manufacturer’s instructions. ABI PRISM 7000 SDS (Applied Biosystems, Foster City, CA, USA) and the commercially available Power SYBR™ Green PCR Master Mix (Cat. No. 4368706; Applied Biosystems, Foster City, CA, USA) were used for RT-qPCR analysis for SNX3, GAPDH (housekeeping gene), and M86 (to confirm MCMV infection).

Oligonucleotide primers for mouse *SNX3* (forward 5′-TGC GGC AGC TTC CTT TTA GA-3′; reverse 5′-AGG ATG ACC AGC GAC CTT GT-3′), *M86* (forward 5′-GGT CGT GGG CAG CTG GTT-3′; reverse 5′-CCT ACA GCA CGG CGG AGA A-3′) [47], and *GAPDH* (forward 5′-CCAATGTGTCCGTCGTGGATCT-3′; reverse 5′-GTTGAAGTCGCAGGAGACAACC-3′) [48] were obtained from Metabion (Planegg, Germany).

In every experiment, SNX3 was normalized using an endogenous control gene (GAPDH). The results are shown as 2^−ΔΔCt^ values [49], calculated as the difference between the Δ^ΔCt^ values of the silenced SNX3 cell line (shSNX3) and the negative control cell line (shScr). The results of the RT-qPCR are shown in comparison to the negative control, which received a value of 1, i.e., 100%.

### 2.11. Plaque Assay and Virus Growth

shSNX3- and shScr-expressing fibroblasts were grown in 24-well plates and the next day were infected with ∆MC95.15. NIH3T3 cells grown in 24-well plates were transfected for 48 h with various siRNAs and then infected with ∆MC95.15. Cell lysates and supernatants were collected 48 h post-infection. The standard plaque assay was used to measure the production of released virions, as previously mentioned [15].

### 2.12. Data Presentation and Statistical Analysis

Statistical analysis was performed using either the Mann–Whitney (U) test or the Kruskal–Wallis test with Dunn’s multiple comparisons by the software MedCalc (version 19.7.2). Statistical significance was determined according to the *p*-value, where the difference was considered significant when the *p*-value was < 0.05 (* *p* < 0.05; ** *p* < 0.01; *** *p* < 0.001; **** *p* < 0.0001). Experiments were performed at least three times, and data are presented as mean ± standard deviation (SD), scatterplot, box-and-whiskers, or a combination of the last two.

## 3. Results

### 3.1. SNX3 Accumulates in the Inner Pre-AC and the Peripheral Membrane System of Infected Cells in the E Phase of Infection and in the AC During the L Phase

The increased recruitment of host cell factors that regulate membrane flux and membrane concentration in the perinuclear region of MCMV-infected cells is a hallmark of the reorganization of the membrane system characteristic of the AC [13]. Therefore, we first investigated whether SNX3 accumulates in the pre-AC during the E phase of infection, which includes the dislocated Golgi and expanded EE-RE/ERC-TGN membranes, and in the fully established perinuclear megastructure of AC after viral DNA replication and MCMV L gene expression. In non-infected NIH3T3 cells, SNX3-positive membrane structures were concentrated in the perinuclear region and at the cell periphery, while the Golgi exhibited typical flattened cisternal structures emanating from the cell center around the nucleus (Figure 1A). In the E phase of infection (6 and 16 hpi), the SNX3-positive structures concentrated in the perinuclear region, and unlinked Golgi cisternae expanded into a ring-like structure, forming the basic configuration of the pre-AC (Figure 1A). The compaction and expansion resulted in increased colocalization of SNX3 and the Golgi marker (Figure 1B), and a considerable proportion of SNX3-positive organelles remained outside the AC at the cell periphery (Figure 1A). The same pattern of SNX3 distribution was also observed in Balb3T3 cells (Appendix A), which provide better resolution for spatial analysis in the later stages of infection. In the L phase of infection, at 48 hpi, the AC was fully established and consisted of expanded Golgi cisternae loaded with the MCMV glycoprotein gpM55 surrounding concentrated SNX3-positive structures, and SNX3-positive structures were reduced at the cell periphery (Appendix A). gpM55 was also present in punctate structures in the inner region that strongly overlapped with SNX3, although the overall colocalization between these two proteins was rather low, as expected (Appendix A).

The enhanced recruitment of SNX3 to the membranes of the inner AC could be due to increased expression of the SNX3 gene, its increased synthesis, or prolonged decay. To analyze this, we performed a Western blot analysis of SNX3 protein expression during the relevant phases of the MCMV replication cycle and found that SNX3 expression was upregulated in the E phase of infection and persisted in the L phase (Figure 1C). Since we did not observe any significant change in SNX3 transcription (Appendix A) when we analyzed our previously published transcriptome of MCMV-infected cells [13], we concluded that SNX3 accumulates in infected cells due to a change in its decay rate. This observation is consistent with the previously published observation of the decrease in EEA1 expression in MCMV- [13] and HCMV-infected [11] cells, as SNX3 competes with EEA1 for binding to PI3P in endosomes [32,35].

In Western blot analysis, we observed two bands stained with anti-SNX3 antibodies, the upper one of about 18 kDa and the lower one of about 15 kDa (Figure 1C). These bands indicate the expression of two SNX3 isoforms in NIH3T3 cells (Figure 2). The 18 kDa band corresponds to the canonical isoform 1 (SNX3-162) of 162 amino acids (aa), while the 15 kDa form possibly corresponds to isoform 2 of 130 aa (SNX3-130) or isoform 4 of 140 aa (SNX3-140) [50]. These protein-coding isoforms are generated by alternative splicing and have been described for both human and mouse SNX3 (Figure 2). The lower molecular weight species accounted for approximately 35% of the total protein detected in the WB. In contrast to the higher molecular weight species, it did not increase during the MCMV replication cycle and accounted for approximately 14–16% of the total SNX protein detected at 16, 24, and 48 hpi (Figure 1D).

### 3.2. SNX3 Localizes at the SNX27-Dependent Rab10-PD

Given that SNX3 is recruited to EEs and may be involved in cargo retrieval in multiple directions, including retrieval of CIE cargo into the CIE cargo recycling route [32], we next investigated whether SNX3 localizes to the same membranes as SNX27 and Rab10, a downstream product of SNX27 activity in MCMV-infected cells [18]. Therefore, we investigated whether SNX3 is associated with SNX27-PD and Rab10-PD by performing a 3D colocalization analysis. We infected stable cell lines expressing inducible EGFP-mSNX27 [18] and EGFP-Rab10, [23] and analyzed SNX3 expression with antibodies against SNX3.

In non-infected cells (not shown) and at 0 hpi (Figure 3A and Appendix A), SNX3 and SNX27 overlapped in restricted areas of the cell, resulting in low colocalization and indicating their distribution at distinct membrane zones.

**Figure 3 biomedicines-13-00936-f003:**
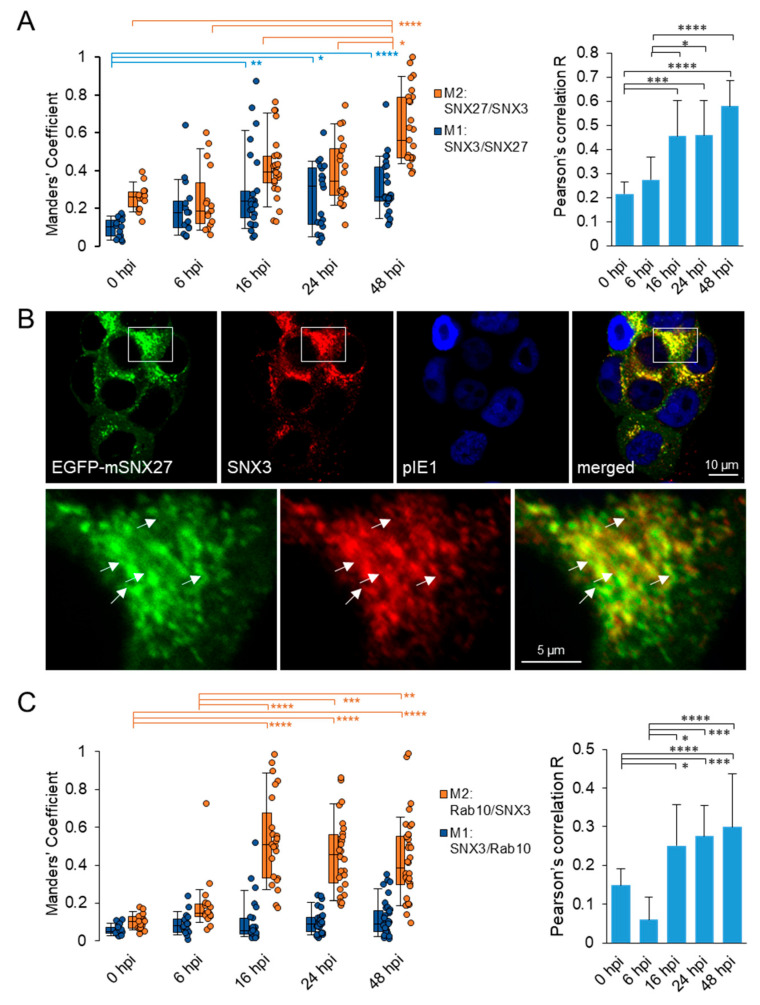
Colocalization analysis of SNX3 with SNX27 and Rab10. (**A**–**C**) NIH3T3 cells with inducible expression of EGFP-mSNX27 or EGFP-Rab10 were treated with doxycycline (2 μg/mL) for 24 h and infected with Δm138-MCMV (MOI of 10). At 0, 6, 16, 24, and 48 h post-infection (hpi), cells were fixed, permeabilized, and stained with primary antibodies against SNX3 and pIE1, followed by appropriate non-cross-reactive secondary antibodies, and analyzed by confocal microscopy. (**A**) Colocalization analysis of EGFP-mSNX27 with SNX3, shown as the percentage overlap (Manders’ coefficients) of SNX3 with SNX27 (M1) and SNX27 with SNX3 (M2), and the Pearson’s correlation coefficient per cell. The measurement was performed over the entire z-stack with the Intermodes-algorithm threshold. The data in the left panel show box-and-whisker plots for M1/M2 and the result for each cell, while the data in the right panel show the mean Pearson’s correlation coefficient for these cells. The error bars show the standard deviation. Representative images of the cells are shown in Appendix A. (**B**) Confocal images of cells at 16 hpi. Shown are the confocal images through the focal plane. The images in the lower panel show the framed area taken at higher magnification. (**C**) Colocalization analysis of EGFP-Rab10 with SNX3, shown as the percentage overlap (Manders’ coefficients) and Pearson’s correlation coefficient per cell, as described in (**A**). Representative images of cells are shown in Appendix A. Statistical significance was determined by the Kruskal–Wallis test with Dunn’s multiple comparisons (* *p* < 0.05; ** *p* < 0.01; *** *p* < 0.001; **** *p* < 0.0001).

At 6 hpi, colocalization of SNX3 with SNX27 already increased (Figure 3A and Appendix A), indicating reorganization of membrane zones in MCMV-infected cells, consistent with the previously observed initiation of membrane system reorganization events [13,16] and enhanced recruitment of SNX27 to membranes within the forming pre-AC [18]. During the E phase, this reorganization continued, and SNX27 was further recruited to membranes of the pre-AC, resulting in the increase in colocalization of SNX27 with SNX3 from 6 to 16 hpi (Figure 3A). It appears that the relocation of SNX3-positive structures to the inner AC was accompanied by its additional enhanced recruitment.

At the end of the E phase of infection, at 16 hpi, SNX27 and SNX3 were concentrated at the same vacuolar structure and likely at the same membrane domains, as colocalization was significantly higher than at the beginning of infection (Figure 3B). Both SNX27 and SNX3 were found at extensions from these vacuolar structures, suggesting their distribution also into tubular domains (Figure 3B, arrows). These vacuolar structures are likely expanded EEs/SEs with enhanced tubulation capacity for generating TRE, consistent with the previously published observation of accumulation and expansion of EEs/SEs within the pre-AC [13,16,39] and the contribution of SNX27 in the generation of TRE [18]. As a substantial proportion of SNX3 remained outside the SNX27-PD (Appendix A), its colocalization with SNX27 increased, although a significant proportion of SNX3-positive membranes remained outside the SNX27-PD, especially at the peripheral membrane system (Appendix A).

With the progression of the replication cycle to the early–late phase (24 hpi), when the pre-AC converted into the AC by loading the L gene product, and to the L phase when the AC was fully developed (48 hpi), the colocalization of SNX27 and SNX3 further increased (Figure 3A), although many membrane domains positive to SNX3 remained outside the area with expanded SNX27-PD (Appendix A). Overall, these data suggest prolonged retention of SNX3 and SNX27 to similar membranes within the AC.

Since retromer–SNX27–ESCPE-1 complexes mediate tubulation of EEs [18] through EHBP1-dependent recruitment of Rab10 [23], we next investigated whether SNX3 remains at Rab10-PD. As expected, very little colocalization of SNX3 with Rab10 was detected at 0 hpi (Figure 3C and Appendix A), but already at 6 hpi, colocalization of Rab10 with SNX3 increased, consistent with the onset of enhanced Rab10 recruitment to the EE domain [23]. Accordingly, Rab10-PD expanded in the perinuclear region up to 16 hpi, as evidenced by the increased recruitment of EGFP-Rab10 in the perinuclear region corresponding to pre-AC (Appendix A), and the recruited Rab10 significantly overlapped with SNX3 (Figure 3C). Nevertheless, a substantial proportion of SNX3 remained outside the Rab10-PD region (Appendix A), resulting in a low degree of colocalization with Rab10. A similar degree of colocalization was observed at 24 and 48 hpi (Figure 3C), at a time when the AC was populated with many L gene products to establish the AC and when the AC was fully developed, respectively. These data, together with the data from colocalization analysis with SNX27, indicate that SNX3 remains associated with membranes producing Rab10-PD.

### 3.3. SNX3 Contributes to the Expansion of Rab10-PD

To further evaluate the contribution of SNX3 to the progression of MCMV infection, we analyzed key events in the MCMV replication cycle in SNX3-depleted cells. We depleted SNX3 by siRNA and by generating cell lines with shRNA (Figure 4A). Both approaches resulted in a significant reduction of SNX3 mRNA as detected by RT-qPCR (Figure 4B).

Western blot analysis showed a significant reduction of 18 kDa SNX3 (average reduction was 73% for non-infected and 48 h infected cells) and moderate or no effect on 15 kDa SNX3 (non-infected average reduction was 23% and for 48 hpi, 3%) in shRNA-treated cells (Figure 4C). The 18 kDa band corresponds to the canonical SNX3-162 isoform (Figure 2). The 15 kDa band does not correspond to the SNX3-140 isoform, as mutation of the Snx3-β2 domain, which the SNX3-140 isoform lacks, structurally impairs its expression, leading to degradation [53]. Furthermore, the 15 kDa band does not correspond to SNX3-113, as this transcript is submitted to the non-sense-mediated decay pathway [52]. Therefore, the 15 kDa band most likely corresponds to SNX3-130, which lacks exon 2 sequences (Figure 2) [50,51,52]. This isoform is probably more resistant to siRNA and shRNA knockdown, as one of the three complementary RNA sequences targets exon 2, which is missing in SNX3-130, and the other two target the 3′ UTR region (Figure 2) [52]. Nevertheless, this isoform appears to be non-functional in EEs because it lacks the key sequences encoded by exon 2 (Figure 2): the β3 sheet, which together with the α1 domain forms the PI3P binding pocket [33], the key amino acids 69–71 required for PI3P binding [32,35], and aa 72, which is required for phosphorylation and termination of PI3P binding [33]. Despite the non-functional phox-homology (PX) domain of SNX3-130 at the EE, this isoform likely retains the ability to act at the cell periphery and PM as it retains retromer binding sequences at the N-terminus [53,56] and other phosphoinositide binding sequences (e.g., PI5P) at the C-terminal stretches [55], as described for the short isoform SNX3-102 which binds to clathrin at the PM [50]. SNX3-102 itself is much shorter, lacking exon 3 and part of exon 4, but contains exon 2 sequences [50] and should therefore be as sensitive to siRNA and shRNA knockdown as SNX3-162. Altogether, the mixture of three siRNA and shRNA sequences resulted in efficient depletion of the canonical SNX3-162 isoform acting at the EEs (Figure 4C).

When shScr- and shSNX3-expressing cells were examined, SNX3 knockdown resulted in reduction of SNX3 (Appendix A, SNX3 staining) but did not result in any observable change in the size and distribution of EEs (Appendix A, Rab5 staining), late endosomes (Appendix A, Lamp1 staining), and the ERC (Appendix A, Rab11a staining), nor did it affect transferrin receptor (TfR) trafficking (Appendix A, TfR). Nonetheless, we observed an increase in EEA1 recruitment to EEs in shSNX3-treated cells (Appendix A), consistent with the reciprocal relationship of SNX3 and EEA1 based on their competition for PI3P and mobilization in different EE zones [32,35]. We also observed a decrease in the recruitment of Vps35 (Appendix A), an essential component of the retromer complex [57], and increased colocalization of Vps35 with EEA1 (Appendix A), consistent with the loss of SNX3-mediated retromer recruitment to EEs and the localization of recruited retromer in EEA1-positive zones of EEs [56]. These data suggest that silencing of 18 kDa SNX3 leads to the expected changes in EE function.

Both knockdowns by siRNA (not shown) and by shRNA had no effect on the establishment of infection, as determined by the expression of pIE1 (Figure 4D), and propagation of the E phase of infection, as determined by flow cytometric quantification of GFP expression in cells infected with recombinant GFP-expressing MCMV (Figure 4E).

Analysis of Rab10-PD expansion in siRNA-treated and shRNA-expressing cells showed that both methods enabled Rab10-PD expansion (Figure 4F). The proportion of cells expressing expanded Rab10-PD was moderately reduced and this reduction was statistically significant, suggesting that SNX3 is not essential for Rab10-PD expansion but contributes to its maintenance.

In contrast to our previously published observations that SNX27–retromer–ESCPE-1-dependent expansion of Rab10-PD contributes to the expression of MCMV proteins [18], depletion of SNX3-associated functions had no effect on the expression of E proteins, as shown by the expression analysis of pM57 and 105 kDa pM25, and of L proteins, as shown by the expression of both forms (130 kDa and 55 kDa) of pM55 and the 70 kDa form of pm74 (Figure 4G). However, we consistently observed a reduction in the expression of L-proteins in shScr-expressing cells, suggesting that this procedure impairs the expression of proteins that depend on DNA replication. Thus, in the comparison between Scr and SNX3 shRNA-expressing cells, the expression of the 55 kDa form of pM55 was significantly increased (Figure 4G). Overall, the analysis of MCMV protein expression suggests that SNX3 depletion does not affect retromer–SNX27–ESCPE-1-dependent sorting into the Rab10 tubular domain, which contributes to the control of viral gene expression. Moreover, it could be that in the depleted SNX3-associated functions, the sorting of the viral gene expression controlling function is even enhanced.

### 3.4. SNX3 Depletion Impairs Virion Assembly but Does Not Reduce Virus Yield

To test whether SNX3 depletion affects virion assembly and egress, we analyzed the intracellular distribution of fluorescent capsids after infection of cells with mCherry-SCP-MCMV. This recombinant virus expresses fluorescently labeled m48.2, a small capsid protein (SCP) that is incorporated into capsids [41]. After infection with this virus, red fluorescence in the form of nuclear condensates accumulated in both Scr and SNX3 shRNA-expressing cells 48 hpi (Figure 4H). The percentage of cells expressing nuclear fluorescence was even higher in SNX3 shRNA-expressing cells. Of the cells expressing nuclear fluorescence, approximately 40% of shScr-expressing cells showed cytoplasmic accumulations of red fluorescence, which is slightly lower than the levels previously observed in non-transfected cells. In cells expressing shSNX3, the percentage of cells exhibiting cytoplasmic mCherry-SCP accumulations was significantly reduced to 30% of cells (Figure 4H). These data suggest that SNX3-associated functions may be required for the formation of cytoplasmic aggregates of fluorescent virions. However, quantification of virions showed no difference in released virions between shScr- and shSNX3-expressing cells, although shSNX3 cells appeared to accumulate significantly more cell-associated virions (Figure 4I,J), suggesting that when SNX3-associated functions are reduced, the cell is able to produce and release infectious virions.

### 3.5. SNX3 Contributes to a Mechanism That Controls Virion Assembly and Egress

To assess the contribution of SNX3 to virion assembly and egress, we knocked down all SNX genes involved in cargo retrieval and membrane bending during the formation of tubular recycling EEs. Their contribution was examined after siRNA silencing in wild-type cells and cell lines expressing shScr or shSNX3, and the formation of infectious virions was monitored using a standard plaque assay as a readout. This approach has the limitation of requiring a lengthy protocol that requires at least two days for suppression of protein levels by transfected siRNA and an additional two to three days for formation and release of infectious virions in infected cells, which may result in dilution of transfected siRNA and partial suppression of associated functions.

Suppression of SNX1 and SNX2, two components of the retromer, SNX4, which is known to be associated with cargo retrieval during endosomal recycling to the PM [58], and SNX17, which is known to contribute to cargo sorting to the TGN [58], had no effect on the detection of released extracellular virions and cell-associated virions (Figure 5A). Combined suppression of SNX1, SNX2, and SNX3 also had no effect (Figure 5A). Consistent with our previously published observation [18], depletion of SNX27 decreased extracellular viral yield, and this effect appeared to be exacerbated by combined depletion of SNX3 (Figure 5A). Similar effects were observed after combined depletion of SNX3 and SNX4 (Figure 5A). These data suggest that SNX3 contributes to the endosomal recycling pathway utilized for virion formation.

As mentioned above, the expression of shRNA for SNX3 had no effect on the production and release of infectious virions. Moreover, the presence of control shRNA consistently reduced virion assembly. Transfection of siRNA for SNX4 and SNX27 to shRNA-expressing control cells significantly reduced extracellular viral yield (Figure 5B). Transfection of siRNA for SNX1, SNX4, SNX17, and SNX27 to SNX3 shRNA-expressing cells significantly reduced extracellular virus yield, and SNX4 also reduced cell-associated virions (Figure 5C). These data suggest that retromer-, SNX4- and SNX27-associated functions contribute to the processes of virion formation and/or release from infected cells.

Overall, the knockdown experiment suggests that cargo sorting processes in the endosomal recycling pathway associated with CIE cargo sorting are required for virion assembly and generation of viral progeny. SNX3-associated functions are not critical, although SNX3 contributes to these processes.

### 3.6. Depletion of SNX3 with shRNA Relocalizes gB

Since SNX3 can contribute to cargo sorting at EEs, its function in MCMV-infected cells can be assumed to be related to the control of viral glycoprotein distribution within the infected cell. To investigate this, we monitored the expression of pM55, which forms the homodimeric complex gB [59], and pm74, known as gO, which forms a complex with gH and gL [59], in shRNA to SNX3-expressing cells. These two glycoproteins have different pathways within the membrane system of the cell. Both accumulate in the outer AC and reach membrane intermediates of the inner AC [13], but only gB is expressed at the cell surface [60,61]. Immunofluorescence analysis of infected cells expressing shRNA for SNX3 showed no remarkable difference in the intracellular distribution of these proteins compared to non-transfected and Scr shRNA-expressing cells (data not shown). However, flow cytometric analysis showed a remarkable increase in gB expression at the cell surface (Figure 6). These data are consistent with the increased expression of the gp55 form of gB (Figure 4), which is generated by proteolytic activity at the cell surface, suggesting that SNX3 regulates the intracellular transport of gB. Analysis of the short-linear motifs (SLIMs) of the gB sequence revealed that gB has SLIMs to recognize the PDZ domain of SNX27 [57] and several sequences that can be recognized by SNX3 (Appendix A).

## 4. Discussion

In this study, we investigated the contribution of SNX3 in the expansion of Rab10-PD, pre-AC biogenesis, and infectious virion production during MCMV infection. The downstream sorting nexins that regulate exit pathways from EEs towards the PM in non-infected NIH3T3 fibroblast-like cells, such as SNX27, transiently and highly dynamically mobilize to EE membranes to generate exit carriers, resulting in their low membrane recruitment [18]. In contrast, SNX3 remains at membranes of steady state EEs and the peripheral membrane system. Its steady-state recruitment to EEs is consistent with its constitutive binding to PI3P-enriched membrane regions of EEs via the PX domain and competition with FYVE domain-containing PI3P-binding proteins such as EEA1 and Hrs [32,36]. Its recruitment to EEs is related to its known functions in retromer-dependent cargo sorting to the TGN [58] and retromer-independent sorting of CIE cargo for recycling to the PM [32], while its recruitment to peripheral membrane organelles is related to its function in clathrin-dependent endocytic processes [50]. In non-infected NIH3T3 cells, two SNX3 isoforms were detected, based on structural SNX3 analyses (Figure 2) [29,33,35,50,51,52,53,54,55,56]. The 18 kDa isoform (SNX3-162) is likely related to the functions of SNX3 in EEs, while the 15 kDa isoform (SNX3-130) is likely related to SNX3 functions on peripheral membrane organelles. Expression of the mixture of three siRNA or shRNA sequences depleted SNX3-162, but not SNX3-130.

In MCMV-infected cells, SNX3-decorated EEs are relocated to the pericentriolar region during the E phase of infection, consistent with their accumulation and expansion within the pre-AC. The expansion is associated with the increase in SNX3-162 expression, which is unrelated to the increase in transcriptional activity, as the SNX3 transcript was not altered during the E phase of infection. These data suggest that CMV infection expands the SNX3-dependent zone of EEs by a mechanism that increases SNX3 protein stability and abundance. The expansion of the SNX3 zone of EEs is associated not only with an increased membrane area for SNX3 recruitment but also with the competition of SNX3 with EEA1, whose recruitment is reduced in both MCMV- [39] and HCMV-infected [11] cells. The observed expansion of the SNX3-positive zone is accompanied by an increased and prolonged recruitment of SNX27 in the same membrane zone, resulting in a strong colocalization of SNX27 with SNX3 and suggesting that SNX27 is recruited to the constitutive SNX3 zone of EEs. This observation suggests that the SNX3-positive domain contributes to sorting CIE cargo for recycling to the PM, as reported in other studies [23], and that MCMV infection specifically expands the SNX3/SNX27 zone for sorting and recycling of the CIE cargo. This zone of EEs is not abundant in non-infected cells, as shown by the low colocalization of SNX3 with SNX27, likely due to its highly dynamic turnover. As infection progressed and the pre-AC matured into the AC, the colocalization of SNX27 with SNX3 further increased, suggesting that the area for recycling CIE cargo within the AC is continuously expanding. Colocalization analysis also revealed a significant proportion of SNX3 outside the SNX27-decorated EE zone, consistent with the SNX3/retromer-dependent EE region for cargo sorting to the TGN and the engagement of SNX3 in the peripheral membrane system [50].

In our previous study, we showed that SNX27, together with retromer (heterotrimer containing Vps26, Vps29, and Vps35) and ESCPE-1 (heterodimeric combinations of SNX1 or SNX2 with SNX5, SNX6, or SNX32), form functional protein complexes at the expanded EE zones within the pre-AC and AC that are required for the progression of the CMV replication cycle [18]. These complexes are known to retrieve CIE cargo and sort it into the CIE cargo recycling pathway for export to the PM [57,58,62]. Cargo sorting is accompanied by the initiation of tubular elongations at the EE membranes, followed by the recruitment of the tubular elongation machinery, elongation of the tubules, and their rapid fission into the recycling carriers [57,58]. Tubular elongation is initiated by the recruitment of EHBP1 to the membranes, accompanied by the activation of Rab10 and the direct binding of Rab10-GTP to EHBP1 [22,63,64]. It is unclear how EHBP1 is recruited to the membrane, but it is likely that the SNX5 component of ESCPE-1 associates with PIPKIγi5, an endosomal variant of PIP5K that generates PI (4,5)P2 [65], and EHBP1 binds directly to PI (4,5)P2 and specifically recruits Rab10-GTP [63]. The recruited Rab10, which binds membranes to the cytoskeleton, recruits motor proteins and other proteins required for membrane elongation and tubule growth [64], which is required for the elongation of membranes into tubules. Tubulation is further supported by Rab10-promoted recruitment of MICAL-L1 [22], a known Rab10 interactor [66], unless MICAL-L1 forms complexes with Pacsin2 and EHD1, which are required for termination of tubulation by cleavage of tubules [22].

In non-infected fibroblasts, the SNX27-associated process appears to occur rapidly, and very little SNX27 and Rab10 can be detected at the membranes [13,15,18,23]. As it is difficult to study such a rapid process, most knowledge about EE-derived TREs has been gained by studying HeLa-M cells that constitutively generate TREs [22,64], likely due to a change in a step required for such dynamics. A similar change in tubulation dynamics was observed in MCMV-infected cells, in both SNX27- [18] and EHBP1/Rab10-dependent [23] steps, resulting in prolonged recruitment of all components to membranes [13,18,23], expansion of SNX27- [14] and Rab10-decorated [15,16,18,23] membrane domains, expansion of vacuolar and tubular domains of EE [13,14,46], retention of CIE cargo [20], and inhibited CIE cargo recycling [20].

Although SNX3 and SNX27 localize to the same EE zone of MCMV-infected cells, SNX3-associated functions are not essential for TRE expansion, as shown by the pericentriolar recruitment of Rab10. Nevertheless, in SNX3-depleted cells, the development of Rab10-PDs and their pericentriolar accumulation is reduced, suggesting that the SNX3-based sorting platform is the rate-limiting step in endosomal tubulation in MCMV-infected cells. In the interaction of SNX3 and SNX27, SNX3 appears to act proximal to SNX27 [34,35], and SNX27 is required to generate tubular intermediates that serve as signaling cues for MCMV gene expression. In the absence of SNX3, the tubular intermediates are generated and MCMV genes are expressed, suggesting that SNX3 is not required for the sorting of host cell factors required for control of MCMV gene expression.

SNX3 could act as an initial sorting mechanism at the EEs to sort cargo away from the degrading region of the EEs [57]. If there is no other cargo sorting signal for sequence-dependent sorting, SNX3 may redirect cargo to the TGN. In the presence of sorting sequences that can recruit SNX27, cargo sorted with SNX3 is redirected to specific regions of the EEs that generate tubular extensions, facilitated by downstream recruitment of Rab10. This is of particular interest for CMV, as viral glycoproteins must either be sorted into tubular EE domains that return cargo to the PM before degradation or redirect glycoproteins to the TGN for recirculation. In the absence of SNX3, a portion of the glycoprotein gB (M55 gene product) is redirected to PM, resulting in an increase in the gp55 form. Like HCMV gB, the M55 gene product is expressed at the PM [60,67], accumulates in the TGN [68], and has an acidic cluster motif required for endocytic uptake [61], although very little information is available about its endocytosis [69]. Like HCMV gB, MCMV gB has multiple sorting motifs that are required for sequence-dependent recycling to the PM via the CIE cargo recycling route and for cycling within the EE-to-TGN route [31,57]. It has a motif that can be recognized by the PDZ domain of SNX27 [18], a group of positively charged residues for ACAP1 binding, and several hydrophobic motifs for ESCPE-1 binding, but no motif for binding with SNX17. It also has several ΦXL-aromatic hydrophobic motifs for recognition by SNX3–retromer and several motifs for binding to AP1 and AP3. In this way, gB can continuously traverse the TGN, PM, EE, and RE loops, forming a typical steady-state pattern with major retention localization in the TGN balanced by SNX27-dependent sorting on the RE–PM route and SNX3–retromer sorting on the dominant RE–TGN route. In the absence of SNX3, the balance of recycling shifts toward the RE–PM loop, resulting in increased exposure at the PM, which in turn leads to increased formation of the gp55 form, which is formed by the proteolytic cleavage of a gp130 kDa precursor at the PM by furin protease activity [70]. Viruses produced in the absence of SNX3 remain infectious, as gB is presumably required for infectivity. The endocytic process does not appear to be impaired and the supply of the gp55 form to the site of the secondary envelopment, which likely occurs in an endocytic compartment [68,71], is sufficient and probably enhanced in SNX3-depleted cells.

SNX3 itself is not required for virion production but may contribute to the control of viral glycoprotein trafficking to the PM recycling region of EEs. This region contributes significantly to MCMV virion production, and depletion of SNX27, SNX4, and possibly SNX17 reduces virion production. These SNXs are key players in the formation of proteinaceous coats and cargo sorting for recycling to the PM [58]. SNX27 acts with retromer and SNX17 acts with Retriever to recycle cargo directly to the PM, while SNX4 acts alone to sort cargo into REs prior to recycling to the PM [58]. SNX3 with retromer redirect cargo to the TGN [58], but also can contribute alone in cargo sorting for recycling to the PM [32]. Therefore, SNX3 and other SNXs, especially SNX27 which can be mobilized by SNX3, may work sequentially along the recycling pathway. The proximal effect of SNX3 in retromer-independent cargo sorting likely creates a scaffold for further recruitment and sorting of cargo by SNX27, SNX4, and SNX17. Here, we show that under conditions of long-term suppression of SNX3 activities using cell lines expressing shRNA for SNX3, MCMV virions are produced and released from the cell. Additional suppression of ESCPE-1 (SNX1), SNX4, SNX17, and SNX27 resulted in reduced release of virions, suggesting that these pathways may be involved in virion biogenesis. The influence of SNX3 is additive, as demonstrated by the simultaneous depletion of SNX3 and SNX27 and by the depletion of SNX27 and SNX4 in cells expressing shSNX3. The additive effects of SNX17 and SNX27 in virion production have been shown for other viruses, such as infection with human papillomavirus [72].

These effects were monitored by detecting the release of infectious virions using the standard plaque assay. This assay indicates the relevant endpoint of virion biogenesis. However, it cannot determine at which step biogenesis is impaired. In our previous study, we showed that suppression of retromer, ESCPE-1, and SNX27 reduced the expression of viral genes [18], suggesting that in addition to the membrane envelopment, the tubular membranes of the CIE cargo recycling pathway of EEs also play a role in regulating the viral replication cycle. The current study showed that these processes do not depend on the activity of SNX3. A major drawback of siRNA experiments in the context of infection is the lengthy procedure associated with dilution of the transfected RNA and adaptation of the cellular membrane system, which is highly redundant. Nevertheless, the siRNA experiments and the monitoring of virion release suggest that several membrane tubulation processes contribute to virion biogenesis.

Although quantification of virion production is not a sensitive tool to study host cell factors that contribute to the lengthy and complex processes of host cell reorganization and virion envelopment, it is the best available test to identify cellular mechanisms that may be involved in these processes. The knockdown experiments in our study indicate that the endosomal recycling system is a host cell system that contributes to virion biogenesis and suggests the involvement of multiple recycling pathways. This observation is consistent with several observations in MCMV- [15,19,73,74] and HCMV-infected [60,61,69,71,75,76,77] cells suggesting virion envelopment at endosomal membranes. However, our recent study has shown that the influence of the endosomal recycling system on virion biogenesis is more complex, as SNX27-dependent endosomal tubulation contributes to the control of viral gene expression [18]. Therefore, a more detailed investigation of endosomal tubulation pathways in CMV-infected cells is required to understand not only the envelopment and egress of CMV but also the complexity of host cell adaptation during infection.

## 5. Conclusions

This study demonstrates that SNX3 contributes to CMV biogenesis and extends our previous studies supporting the role of EE- and RE-derived membrane structures in CMV replication, the final stages of virion assembly, and egress from the cell [15,18,23,46]. Attempts to decipher the role of host cell factors shaping EE/RE zones and domains using siRNA and shRNA tools in such long processes encounter redundancy of cellular processes. This is also evident in this study, which suggests that CMV assembly pathways, both final envelopment and egress organelle assembly, may be constructed by available membrane types within the large area of the inner AC available at the time of these events, rather than by a single and specific type of membrane domain. This is similar to the recently described process of octopus-like engulfment of autophagic material by EE/RE-derived membrane extensions [78]. These processes could be related to the heterogeneity of the host cell response to infection and the stochastic selection of available membranes at the time of envelopment. This stochastic nature could also explain the heterogeneity of the observed host cell factors, but also of the viral glycoproteins within the released virions identified in recent high-throughput studies [74,76,79,80,81,82,83,84]. This heterogeneity could represent a further obstacle to high-throughput analyses, even at the single-cell level. Therefore, it is crucial to further and more deeply understand the composition of the membranes within the AC, including the SNX3-decorated EE zones investigated in this study, to obtain a comprehensive picture of how CMV establishes the egress route. Once this picture is established, developing a method to demonstrate its existence in CMV-infected cells becomes even more challenging. Thus, the present study makes a small contribution to the complex puzzle of CMV biogenesis.

## Figures and Tables

**Figure 1 biomedicines-13-00936-f001:**
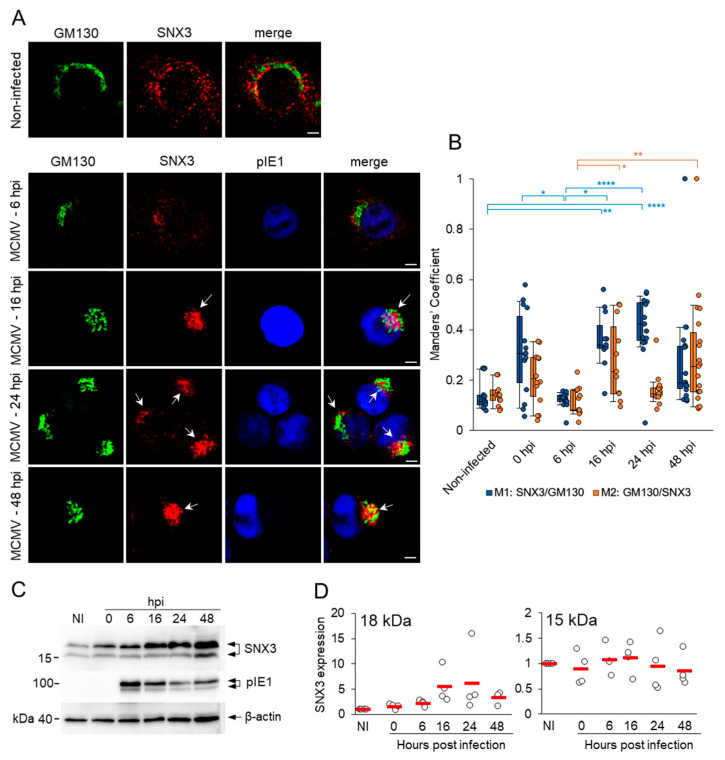
Perinuclear accumulation of SNX3 in the pre-AC and AC of MCMV-infected cells. (**A**) Immunofluorescence analysis of SNX3 and GM130. NIH3T3 cells were infected with Δm138-MCMV (MOI 10) or left non-infected, fixed at 6, 16, 24, and 48 hpi, permeabilized, and stained with Abs against SNX3 (red) in combination with Abs against GM130 (green) and pIE1 to control infection (blue). Shown are confocal images through the focal plane of a representative experiment. The arrows indicate perinuclear accumulation in the pre-AC. Bars, 5 μm. (**B**) Colocalization analysis shown as the percentage overlap (Manders’ coefficients) of SNX3 with GM130 (M1) and GM130 with SNX3 (M2) per cell. The measurement was performed over the entire z-stack with the Intermodes-algorithm threshold. The data panel shows box-and-whisker plots for M1/M2 and the result for each cell from three independent experiments. (**C**) Western blot analysis of SNX3 expression during the E (6 and 16 hpi) and L (24 and 48 hpi) phases of infection. The expression of pIE1 and β-actin in each sample served as the infection and loading controls, respectively, and were performed on the same membranes (Appendix A). (**D**) Quantification of 18 kDa and 15 kDa SNX3 signals using ImageQuantTL software (version 10.2). The expression was normalized to non-infected cells. Shown are the individual results (empty circles) and the average (red bars) of three to four independent experiments (shown in Appendix A). Statistical significance was determined by the Kruskal–Wallis test with Dunn’s multiple comparisons (* *p* < 0.05; ** *p* < 0.01; **** *p* < 0.0001).

**Figure 2 biomedicines-13-00936-f002:**
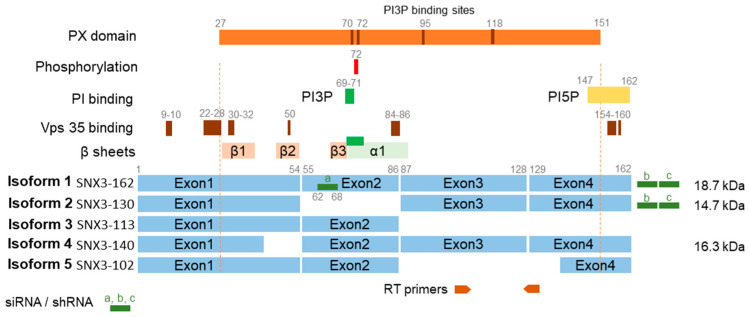
Transcript isoforms of SNX3 (related to Figure 1C and Figure 3). Five isoforms of SNX3 can be generated by alternative splicing [50,51]. Protein-coding isoform 1 is generated by four exons and encodes a protein of 162 amino acids (aa), which is known as a canonical protein of 18.7 kDa (SNX3-162) [51]. Protein-encoding isoform 2 is generated by the exclusion of exon 2 and encodes a protein of 130 aa (SNX3-130) and 14.7 kDa, while protein-encoding isoform 4 is generated by truncation of exon 1 and encodes a protein of 140 aa (SNX3-140) and 16.3 kDa [51]. All three isoforms are found in human and mouse cells [51]. Isoform 3 lacks exons 3 and 4 and is predicted to undergo nonsense-mediated decay [52]. Isoform 5 is protein-coding and has been described in mouse cells [50]. It lacks exon 3 and part of exon 4 and encodes a protein of 102 aa (SNX3-102) that cannot bind to PI3P at EEs, but has the ability to bind to clathrin at the PM and peripheral endosomal system [50]. An important functional domain of SNX3 is the phox-homology (PX) domain, which spans position 27–151 and contains three β-sheets (β1-3) that are essential for SNX3 function [29,33,53,54]. Mutation of β2 impairs the structure of the protein and leads to its degradation, while mutations of β1 and β3 lead to loss of SNX3 function in vivo [53]. Association with endosomes is mediated by PI3P binding via the PX domain at four essential sites (brown bars in the schematic representation of the PX domain), but mutagenesis studies have shown that position 69–71 is critical (green) [35]. Sequences 147–162 at the carboxy terminus allow preferential binding to PI5P [55]. Mutagenesis studies have revealed several essential points in the SNX3 structure for binding of the Vps35 component of retromer (brown bars) [53,56]. Based on these structural features of SNX3, the protein-coding isoform 2 (SNX3-130) should be expressed but lose the ability to bind to PI3P of EEs, similar to isoform 5 (SNX3-102) [50], and retain the ability to bind to membranes other than EEs (e.g., PM). Isoform 4 should be depleted as the β2-sheet is essential for the stability of SNX3 [53]. The positions of the primers used for RT-qPCR in this study are shown below, and the positions of the siRNA and shRNA reagents found in transcript sequences are indicated by green boxes. The numbers indicate the amino acid boundaries and positions.

**Figure 4 biomedicines-13-00936-f004:**
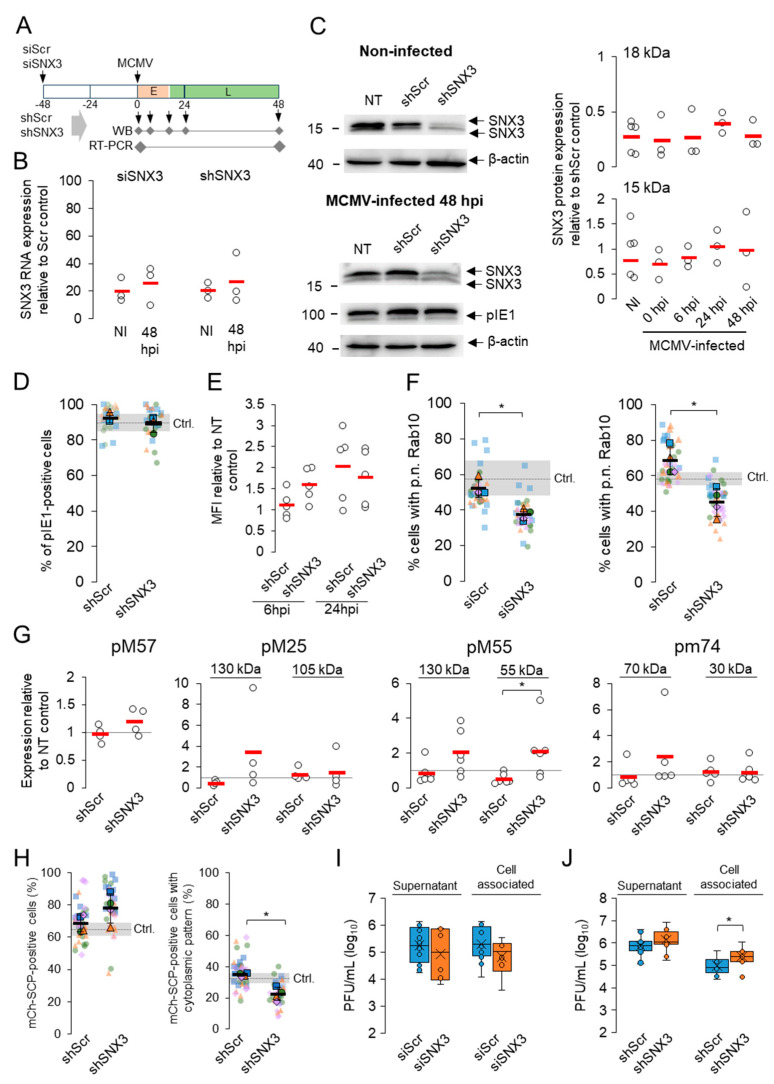
Effects of SNX3 knockdown on MCMV replication. (**A**) Schematic representation of the knockdown experiments. NIH3T3 cells were either used as cell lines expressing shScr and shSNX3 or transfected with siScr and siSNX3 (80 nM) 48 h prior to infection. Non-transfected (NT) cells were used as a control. Cells were infected with MCMV (MOI 10), and SNX3 knockdown was tested by RT-qPCR (48 hpi) and Western blot at 0, 6, 24, and 48 hpi. Infected cells were examined at early (0, 6, and 16 hpi) and late (24 and 48) stages of infection for expression of SNX3, viral proteins, the establishment of the pre-AC and nuclear replication centers, expression of MCMV capsids, and production of infectious virions. (**B**) RT-qPCR quantification of SNX3 expression in non-infected and 48 h-infected siSNX3-treated and shSNX3-expressing cells. Data represent three independent experiments normalized to the shScr control. The red bar shows the mean value. (**C**) Representative Western blots showing SNX3 depletion in non-infected and Δm138-MCMV-infected (48 hpi) cells. Expression of pIE1, as a control for infection, and β-actin, as a loading control, were determined on the same blotting membranes. Data on the right represent individual blots normalized to β-actin expression in each sample, followed by normalization to the shScr. The red bars show mean values. (**D**) The percentage of shScr- and shSNX3-expressing cells with nuclear staining of pIE1 at 16 hpi as determined by immunofluorescence. Data show individual experiments, mean values (black bar), and SD. Control (Ctrl.) line represents the mean ± SD in non-transfected cells. (**E**) Expression of GFP was determined by flow cytometry after infection with GFP-MCMV. (**F**) Percentage of infected (Δm138-MCMV, pIE1-positive) siScr- and siSNX3-treated (left graph) or shScr- and shSNX3-expressing (right graph) cells with perinuclear (p.n.) accumulation of Rab10. Data represent individual experiments, mean values (black bar), and SD. Ctrl. represents mean values ± SD of Rab10 accumulation in non-transfected cells. (**G**) Expression of the viral proteins pM57, pM25 (130 and 105 kDa form), pM55 (130 and 55 kDa form), and pm74 (70 and 30 kDa form) were determined by Western blot analysis in shScr- and shSNX3-expressing cells at 48 hpi (Δm138-MCMV). Data represent individual blots normalized to β-actin expression in each sample, followed by normalization to the non-transfected (NT) control. The red bars show mean values. Blots of all experiments are shown in Appendix A. (**H**) Percentage of shScr- and shSNX3-expressing cells developing SCP accumulation after infection with S-SCP-mCherry-MCMV at 48 hpi (left panel). The right panel shows the percentage of SCP-expressing cells that develop cytoplasmic SCP accumulation. Data represent quantification by immunofluorescence of different fields from four independent experiments. Shown are the mean values (black bars) and the average values of the individual experiments. Ctrl. stands for mean values ± SD in the non-transfected control. (**I**,**J**) Quantification of released and cell-associated infectious virions in siScr- and siSNX3-treated cells, (**I**) or shScr- and shSNX3-expressing cells, (**J**) determined by the plaque assay 48 h after infection with Δm138-MCMV. Box-and-whisker plots represent data from 6–12 individual experiments with shown inner points and outliers. Statistical significance was determined using the Mann–Whitney test (* *p* < 0.05).

**Figure 5 biomedicines-13-00936-f005:**
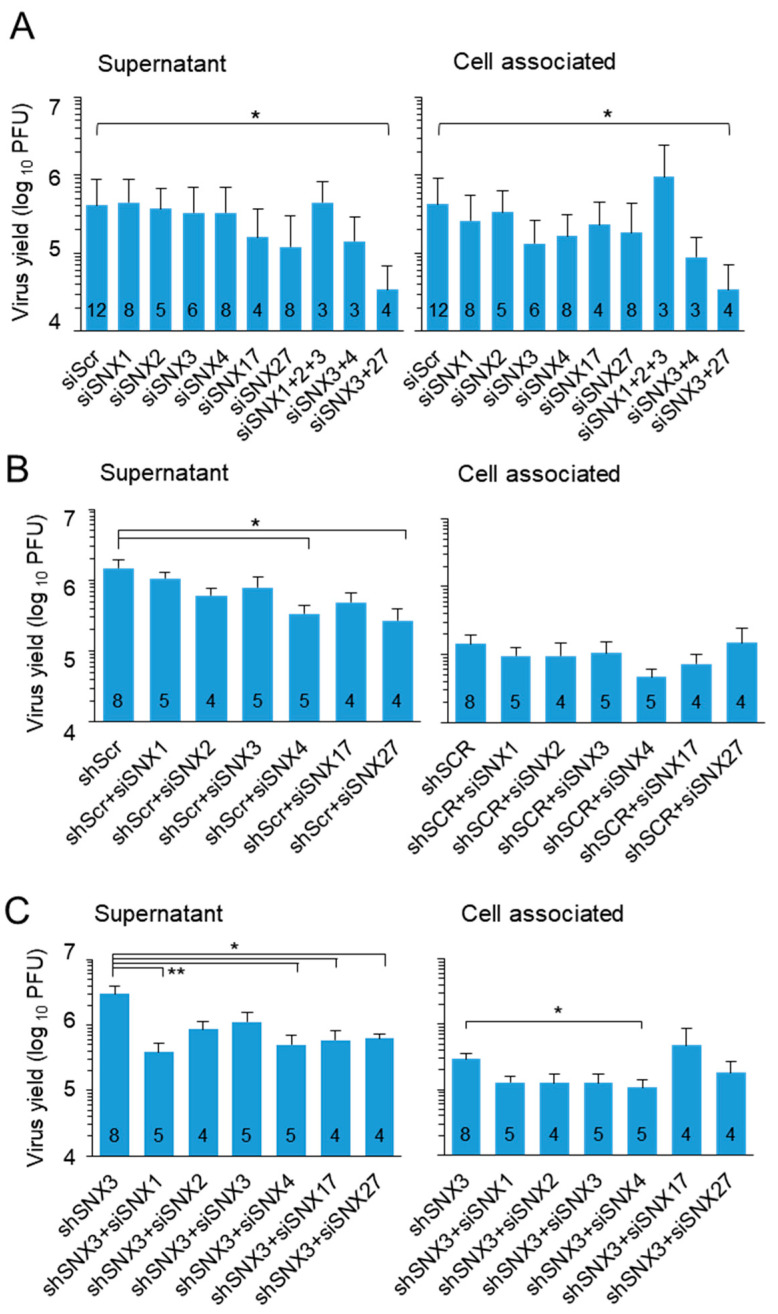
Knockdown of sorting nexins associated with endosomal recycling to the PM confirms the contribution of SNX3 to MCMV virion biogenesis. Knockdown of SNXs by siRNA in (**A**) NIH3T3 cells, (**B**) NIH3T3 cells expressing shScr, and (**C**) NIH3T3 cells expressing shSNX3. Cells were treated with siRNA for 48 h, infected with Δm138-MCMV (MOI 10), and the amounts of cell-associated and released (supernatant) infectious particles were determined 48 hpi using a standard plaque assay. Data are presented in log10 scale and represent the mean and standard deviation of 3–12 experiments (the number of experiments is indicated in the bars). Statistical significance was determined by the Kruskal–Wallis test with Dunn’s multiple comparisons (* *p* < 0.05; ** *p* < 0.01).

**Figure 6 biomedicines-13-00936-f006:**
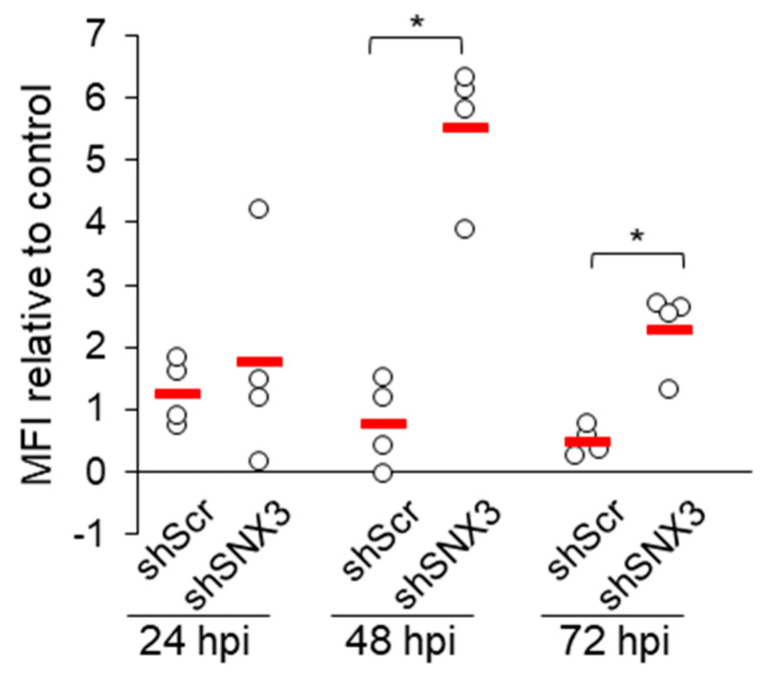
Knockdown of SNX3 increases the cell surface level of pM55 (gB). NIH3T3 cells expressing shScr or shSNX3 were infected with Δm138-MCMV (MOI 10), and the expression of pM55 was determined at 24, 48, and 72 hpi using flow cytometry. Data represent ΔMFI of independent experiments normalized to non-transfected cells, and the red bars show mean values. Statistical significance was determined using the Mann–Whitney test (* *p* < 0.05).

## Data Availability

The data presented in this study are available on request from the corresponding author.

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
