# Peer review of "Contribution of Sorting Nexin 3 in the Cytomegalovirus Assembly"

_biomedicines, 2025, doi:10.3390/biomedicines13040936_

Round 1
Reviewer 1 Report
Comments and Suggestions for Authors
The manuscript entitled "Contribution of sorting nexin in the cytomegalovirus assembly". Title, abstract and overall rationale of work is satisfactory. However, there are major concerns, which needs to be addressed before publication.
1) The overall the abstract written well but conclusion part should be modify and write more informatics.
2) In the introduction section: Author do not repeat abbreviation again and again. Secondly, author wrote more details about the CMV and very less describe about the SNXs. Pre-AC. I suggest author need to describe about these as well as. Moreover, author need to write concise way with more informatics.
3) Material and methods section: Section 2.2: author need to write details about Virus stocks were prepared according to standard procedures. Please write details how to prepare. Other material and methods part is written well.
4) Results section: In the text figure 1 B is not mention please mention in appropriate place. In the figure legend author need to write how many times they repeated the experiment. Secondly, I observed author wrote more details in legend compare to result section. Overall results section written well and describe organized way.
5) Discussion part: This section author need to improve because author written the results part but they do not discuss properly and I saw the lack of discussion in this manuscript. I recommend author, they should elaborate the discussion part and author need to revise and compare the study with relevant study.
6) Some references are too old (ref 23, 36 and 50) and author need to revise if study is available.
Author Response
Thank you for taking the time to review this manuscript. You can find the detailed responses below and the corresponding revisions in the Track Changes in the newly submitted files. We have also reorganized the order of references, and these interventions are not indicated by track changes. New references are marked in red.
Comment 1: The overall the abstract written well but conclusion part should be modify and write more informatics.
Response 1: Thank you for pointing this out. We agree with this comment and therefore the conclusion part of the abstract has been changed (lines 32-37 of the revised manuscript with track changes). Now, we believe, is more informative.
Comment 2: In the introduction section: Author do not repeat abbreviation again and again. Secondly, author wrote more details about the CMV and very less describe about the SNXs. Pre-AC. I suggest author need to describe about these as well as. Moreover, author need to write concise way with more informatics.
Response 2. I agree with the comment. We think that the addition of the paragraph on sorting nexins would improve the Introduction and, more importantly, significantly support the Discussion, which was somewhat missing. Therefore, the Introduction section has been revised to be more concise and provide the reader with the necessary information (lines 64-119 of the revised manuscript with track changes). Abbreviations that are used no more than three times have been removed. The paragraph focusing on sorting nexins has been included. This paragraph mainly focuses on the role of SNX in endosomal recycling, as this is the main focus of the manuscript (lines 132-164 of the revised manuscript with track changes). The explanation of the role of SNX in recycling processes provides explanations for the discussion.
Comment 3: Material and methods section: Section 2.2: author need to write details about Virus stocks were prepared according to standard procedures. Please write details how to prepare. Other material and methods part is written well.
Agree. The details on the preparation of virus stocks have been included (lines 188-196 of the revised manuscript with track changes)
Comment 4. Results section: In the text figure 1 B is not mention please mention in appropriate place. In the figure legend author need to write how many times they repeated the experiment. Secondly, I observed author wrote more details in legend compare to result section. Overall results section written well and describe organized way.
Figure 1B is mentioned at an appropriate place (lines 384-385 of the revised manuscript with track changes) and the number of experiments is given in the figure legend (line 405 of the revised manuscript with track changes).
Comment 5: Discussion part: This section author need to improve because author written the results part but they do not discuss properly and I saw the lack of discussion in this manuscript. I recommend author, they should elaborate the discussion part and author need to revise and compare the study with relevant study.
Response 5: Agree only in part. We have intervened in the discussion section to improve it (marked with track changes). We found that the insertion of the paragraph on sorting nexins was beneficial for discussion and provided much-needed explanation for the discussion. We found that the discussion is extensive and covers all that is known about SNX3 in the context of events within the AC of MCMV-infected cells. The only relevant study is the excellent paper by Tian et al. (2021), which is extensively discussed. There is no study in CMV research to compare the data presented in the manuscript.
Comment 6: Some references are too old (ref 23, 36 and 50) and author need to revise if study is available.
Response 6: We disagree to exclude mentioned references. These references are rather old, but not too old. They are seminal for Rab10, SNX3, and gB. Therefore, they cannot be avoided. The reference by Babey et al. (2006) is the first reference describing the cell biology and imaging of Rab10 in living cells. None of the subsequent references contain such data, and the inclusion of some more recent reviews would not be correct. The same applies to the reference on SNX3 Xu et al. (2001), which is still up-to-date in several respects and should be cited. The two very recent references by the lead author of the Wu et al. (2001) study, one of which is a review, are included but do not cover the experimental information contained in this study. Tugizov et al. (1999) published the acidic cluster of gB, which is required for endocytic uptake and cannot be replaced without violating the intellectual property.
Reviewer 2 Report
Comments and Suggestions for Authors
Review of Biomedicines manuscript -3521701
This manuscript investigates the potential role of sorting nexin SNX3 in the membrane remodelling which is a part of the cellular response to CMV infection and is essential for virus formation and release.
The model system used is the murine CMV as this allows for simpler examination of earlier cellular events in infection.
The manuscript describes a considerable amount of data on the localisation and changes in localisation of SNX3 in relation to other membrane compartment markers in response to viral infection. This is achieved mainly by using quantitative immune fluorescence microscopy, combined with specific modulation of protein expression by siRNA and shRNA.
The main findings are that a specific isoform of SNX3, namely SNX3-162, contributes to but is not essential for virus assembly and release.
This is a significant finding and in the words of the authors “makes a small contribution to the complex puzzle of CMV biogenesis”.
The manuscript is well written, and the data well presented, though the figures are a little data packed.
Experimental methodology is appropriate to the study and experiments well described.
There are a few issues that need to be addressed before publication.
- Figures use “Hours post infection” as a marker for stage of infection. In the text the authors also talk about early and late stages of infection (see line 369 for example). The authors should define which time points they consider to be in which stage of infection, where does the switch from early to late occur in the 48hr time scale.
- In Fig 3 panel B. Arrows are used to indicate tubular structures in the separate panels for EGFP-mSNX27 and SNX3. These seem to indicate separate regions on the two images and are not included on the merged image. Inclusion of the arrows on the merged image would allow a better judgment of these structures contain both proteins or not.
- Section 3.3 lines 488 – 490 discuss the effect of expression knockdown by siRNA and shRNA. Effects are described as “moderate” and “significant” an actual quantitative assessment should be provided in terms of % knockdown compared to native.
- Fig S8 – labelling of top two panels is missing. I assume these are A and B.
Author Response
Thank you for taking the time to review this manuscript. You can find the detailed responses below and the corresponding revisions in the Track Changes in the newly submitted files. We have also reorganized the order of references, and these interventions are not indicated by track changes. New references are marked in red.
Comment 1: Figures use “Hours post infection” as a marker for stage of infection. In the text the authors also talk about early and late stages of infection (see line 369 for example). The authors should define which time points they consider to be in which stage of infection, where does the switch from early to late occur in the 48hr time scale.
Response 1: Thank you for pointing this out. We agree. We have intervened in the introduction section (lines 79-83 of the revised manuscript with tracking changes).
Comment 2: In Fig 3 panel B. Arrows are used to indicate tubular structures in the separate panels for EGFP-mSNX27 and SNX3. These seem to indicate separate regions on the two images and are not included on the merged image. Inclusion of the arrows on the merged image would allow a better judgment of these structures contain both proteins or not.
Response 2: Agree. The arrows are included in the merged images.
Comment 3. Section 3.3 lines 488 – 490 discuss the effect of expression knockdown by siRNA and shRNA. Effects are described as “moderate” and “significant” an actual quantitative assessment should be provided in terms of % knockdown compared to native.
Response 3: The quantitative data have been inserted into the text (lines 538-540 of the revised manuscript with track changes).
Comment 4: Fig S8 – labelling of top two panels is missing. I assume these are A and B.
Response 4: The top panels are labeled with A and B in Figure S8.
Reviewer 3 Report
Comments and Suggestions for Authors
Major comments:
Line 182: Why were 60%-70% confluent cells used for seeding? Similarly, an MOI of 10 was used for infection. Authors should provide reason and references for the selection of these values.
Line 365-374 and Figure S4: I suggest the authors provide RT-qPCR data to evaluate the SNX3 transcription and to conclude that SNX3 accumulates in infected cells due to a change in its decay rate.
Minor comment:
Line 605: Authors should add all the genes knocked down and not use “SNX3 and other SNXs involved."
Author Response
Thank you for taking the time to review this manuscript. You can find the detailed responses below and the corresponding revisions in the Track Changes in the newly submitted files. We have also reorganized the order of references, and these interventions are not indicated by track changes. New references are marked in red.
Comment 1: Line 182: Why were 60%-70% confluent cells used for seeding? Similarly, an MOI of 10 was used for infection. Authors should provide reason and references for the selection of these values.
Response 1. The propagation of cells to 60-70% confluence was used to avoid heterogeneous differentiation of fibroblast-like cells NIH3T3. This was based on many years of experience in our laboratory and was recently explained in the paper by Rahimi et al. (2022) (line 232 of the revised manuscript with track changes). The reference is included. The MOI of 10 was used to achieve infection of all cells. This MOI is required to achieve at least 96% of IE1-expressing cells, which we regularly determine as a control for infection in our experiments and explained in our publications Karleuša et al. (2018) and Lucin et al. (2021). These references are included (lines 200-201 of the revised manuscript with track changes)
Comment 2: Line 365-374 and Figure S4: I suggest the authors provide RT-qPCR data to evaluate the SNX3 transcription and to conclude that SNX3 accumulates in infected cells due to a change in its decay rate.
Response 2: We observed no significant difference in SNX3 expression in the transcriptome analyses and no significant change during RT-qPCR assays of SNX3 RNA expression for knockdown experiments. We considered that is sufficient to include the transcriptome data at 2 and 18 hpi (Figure S4) as we considered this information more appropriate. Therefore, the accumulation of SNX3 is likely associated with the prolonged lifespan of the SNX3 protein. This is not surprising as CMV alters the expression levels of many proteins, as demonstrated by Weekes et al. (2014) in HCMV-infected cells. This proteomic analysis showed that SNX3 also accumulates in HCMV-infected cells.
Comment 3: Line 605: Authors should add all the genes knocked down and not use “SNX3 and other SNXs involved."
Response 3: We have intervened in the text (lines 656-658 of the revised manuscript with track changes)
Round 2
Reviewer 1 Report
Comments and Suggestions for Authors
The authors have addressed all the concerns raised in the previous version of the manuscript and the quality has much improved after incorporating required modifications. Therefore, the manuscript may be considered for publication in this Journal.
Reviewer 3 Report
Comments and Suggestions for Authors
The revision to the manuscript has greatly improved its clarity and depth. I commend the authors for their hard work in responding to the feedback. I recommend that the manuscript be accepted for publication.